# Functional but not obligatory link between microsaccades and neural modulation by covert spatial attention

Baiwei Liu [1✉], Anna C. Nobre [2,3] & Freek van Ede [1,3✉]

Covert spatial attention is associated with spatial modulation of neural activity as well as with directional biases in fixational eye movements known as microsaccades. We studied how these two 'fingerprints' of attention are interrelated in humans. We investigated spatial modulation of 8-12 Hz EEG alpha activity and microsaccades when attention is directed internally within the spatial layout of visual working memory. Consistent with a common origin, spatial modulations of alpha activity and microsaccades co-vary: alpha lateralisation is stronger in trials with microsaccades toward versus away from the memorised location of the to-be-attended item and occurs earlier in trials with earlier microsaccades toward this item. Critically, however, trials without attention-driven microsaccades nevertheless show clear spatial modulation of alpha activity – comparable to trials with attention-driven microsaccades. Thus, directional biases in microsaccades correlate with neural signatures of spatial attention, but they are not necessary for neural modulation by spatial attention to be manifest.

[1] Institute for Brain and Behavior Amsterdam, Department of Experimental and Applied Psychology, Vrije Universiteit Amsterdam, Amsterdam, The Netherlands. [2] Department of Experimental Psychology, University of Oxford, Oxford, UK. [3] Oxford Centre for Human Brain Activity, Wellcome Centre for Integrative Neuroimaging, Department of Psychiatry, University of Oxford, Oxford, UK. ✉email: b.liu@vu.nl; freek.van.ede@vu.nl

C overt attention allows us to select and prioritise relevant sensory information that is not currently fixated, such as peripheral objects in the external environment or internal object representations in working memory[1–4]. It is well known that the brain's oculomotor system that controls eye movements also participates in the allocation of covert spatial attention[5–9]. One particularly striking demonstration of this is the directional biasing of fixational eye movements—known as microsaccades[10–12]—during the deployment of covert spatial attention[13–20]. This occurs not only when directing attention to external locations[13–16,20], but also when directing attention to internal representations held within the spatial layout of working memory[17–19].

In studying the neural basis of covert spatial attention, the potential contribution of microsaccades is easily overlooked because these eye movements fall within threshold boundaries for what counts as maintaining fixation (i.e., typically occurring within 1 degree visual angle). Yet, given the consistent link between covert spatial attention and microsaccades, it is important to delineate how microsaccadic and neural signatures of covert spatial attention are related[10,21–25]. A recent study in non-human primates by Lowet et al.[21] concluded that enhanced neural processing by covert spatial attention (as indexed by spatially specific firing-rate increases in V4) occurs only in the presence of microsaccades toward the attended location. This argues for an obligatory link between microsaccadic and neural signatures of covert spatial attention.

Here we revisited the nature of this link between microsaccades and modulation of neural activity by covert spatial attention in humans. To this end, we investigated spatial modulation of 8–12 Hz alpha activity—a canonical neural signature of spatial attention in human EEG and MEG measurements[26–30] that has been linked to the oculomotor system[31–33]. Critically, while the links between covert spatial attention and spatially specific alpha modulation[27–30] as well as directional microsaccade biasing[15,20] have each been known to exist for approximately 20 years to date, little remains known about how these two separate hallmark signatures of covert spatial attention are related.

We address this question in a context in which participants directed attention to internal visual representations held within the spatial layout of working memory. This provided a good model system for several reasons. First, such 'internal selective attention' is associated with robust microsaccade biases[17–19] as well as with clear alpha lateralisation[34–40]. Second, unlike when (covertly) selecting peripheral visual objects, when selecting internal visual representations in working memory there is no incentive (functional utility) for overt gaze behaviour as there is no external object to look at. In this context, directional biases in microsaccades are a relatively pure reflection of covert attentional processes. Finally, potential differences in neural activity by attention in this context cannot be due to retinal effects associated with the attended stimulus moving closer to the fovea, as no stimulus is physically present.

To preview our results, we confirm that neural modulation and directional biases in microsaccades are co-modulated by spatial attention: attentional alpha lateralisation is stronger in trials with microsaccades toward vs. away from the to-be-attended item and occurs earlier in trials with earlier microsaccades toward this item (in agreement with ref. [21]). At the same time, however, our data bring the important and nuancing insight that trials with no discernible microsaccade (in the attentional selection window of interest) nevertheless show clear alpha lateralisation (comparable to the lateralisation observed in trials with attention-driven microsaccades). Thus, while directional biases in microsaccades are correlated with neural signatures of spatial attention, they are not necessary for neural modulation by covert spatial attention to occur.

## Results

Twenty-five healthy human volunteers performed a covert spatial attention task requiring them to select one of two visual item representations maintained in working memory. Participants remembered two coloured bars with different orientations. At encoding, one always appeared on the left and one on the right. After a short memory delay, the fixation cross changed colour acting as the attentional selection cue. Participants reported the orientation of the colour-matching memory item using a central reporting dial (Fig. 1a). In this set-up, there is no need or incentive for making eye movements. The relevant sensory information exists only as an internal memory representation. Furthermore, both the selection cue and the reporting dial appear centrally, avoiding any sensory-driven spatial attention shifts.

In separate articles (addressing distinct questions using a common dataset), we recently reported that spatial attention in this task is associated with spatial modulation of microsaccades[17] as well as with spatial modulation of posterior 8–12 Hz alpha activity[35]. Here we uniquely investigated how these two signatures are related.

**Microsaccades are directionally biased during internal spatial attention in many, but not all, trials.** Building on our prior demonstration of a 'gaze bias' during internal selective attention[17–19], the selection cue triggered clear directional biases in gaze shifts (Fig. 1b; for our gaze-shift detection pipeline see Supplementary Fig. 1). From ~200–600 ms after cue-onset, we observed significantly more gaze shifts toward vs. away from the memorised location of the cued memory item (horizontal black line in Fig. 1b; cluster $P < 0.001$). Figure 1c shows this directional bias as a function of gaze-shift magnitude, revealing that this bias is almost completely driven by small gaze shifts in the microsaccade range. Whereas items in the memory display were placed at 5.7° (corresponding to 100% in our analysis), the attentional modulation was driven almost exclusively by gaze shifts smaller than 1°. This was also clear when we considered gaze-shift rate as a function of shift magnitude separately for toward and away gaze shifts (Supplementary Fig. 2). These data thus confirm a robust microsaccade bias during internally directed spatial attention.

To guide subsequent behavioural and EEG analyses, we defined the period of 200–600 ms post-cue as the 'attentional selection period of interest' based on the gaze-bias time course. We then sorted all trials with a usable eye trace in this time period into three types of trials (Fig. 1d): (1) trials in which the first-detected microsaccade in this 200–600 ms period was directed *toward* the memorised location of the to-be-attended memory item ("toward"), (2) trials in which the first-detected microsaccade was directed in the opposite direction ("away"), and (3) trials in which *no* discernible microsaccade was detected in this 200–600 selection period of interest. This revealed two things. First, as expected, this confirmed a clear dominance of toward vs. away microsaccades and showed how this was the case in each participant (Fig. 1d; see also Supplementary Fig. 3). On average, we found $44 \pm 2\%$ ($M \pm SE$) "toward" and $22 \pm 1\%$ "away" trials per participant (corresponding to $350 \pm 22$ and $175 \pm 9.7$ trials per participant, respectively). Second, this also revealed a considerable number of trials with no detected microsaccades in this selection period of interest ($34 \pm 3\%$, corresponding to $271 \pm 23$ trials per participant). This enabled us to separately examine behavioural performance and neural modulation by spatial attention in these three classes of trials—to which we turn next.

**Microsaccade direction predicts reproduction performance for the cued memory item.** Having separated our trials into toward, away, and no-microsaccade trials, we first looked for potential

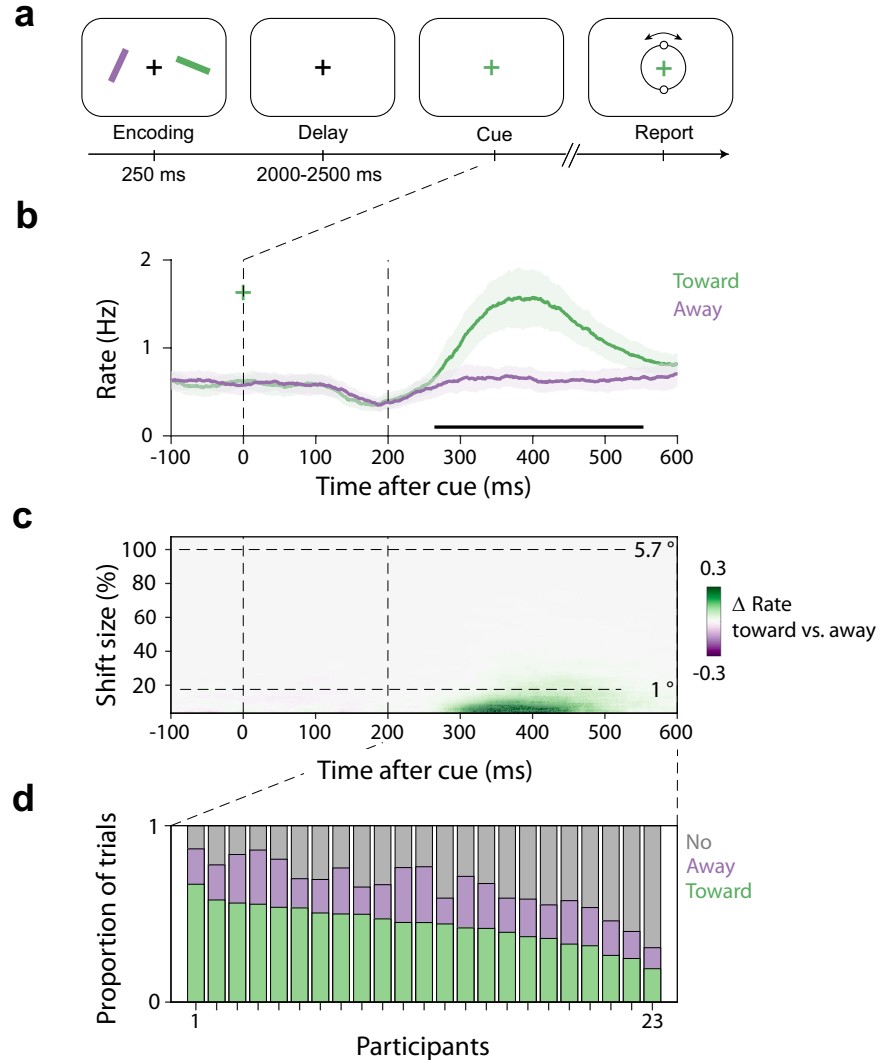

**Fig. 1 Directional biases in microsaccades during internal spatial attention following a non-spatial colour cue. a** Task schematic. Participants memorised two orientated bars with different colours to reproduce the orientation of one bar stored in working memory after a short delay. Following the delay, a colour change of the central fixation cross served as the attentional cue that prompted participants to select and report the orientation of the colour-matching item from memory. After response initiation, a central response dial appeared on the screen and participants performed a reproduction report. **b** Time courses of gaze shift rates (number of microsaccades per second) for shifts toward and away from the memorised location of the cued memory item. Black horizontal line indicates the significant temporal cluster (cluster-based permutation test). Time courses show mean values, with shading indicating 95% CI (calculated across 23 participants). **c** Difference in gaze-shift rates (toward minus away; see Supplementary Fig. 2 for towards and away separately) as a function of gaze-shift magnitude. For reference, dashed horizontal lines indicate 1° visual angle, as well as the original location of the memory item (100% in our analysis, corresponding to 5.7°). **d** The relative proportion of toward, away, and no-microsaccade trials in the 200–600 selection window, as a function of participant. For visualization purposes, we sorted participants by their proportion of toward-microsaccade trials.

behavioural differences associated with the three microsaccade classes.

When considering the accuracy of the dial-up reproduction report (Fig. 2a), we found a significant main effect of microsaccade class ($F(2, 44) = 3.39$, $P = 0.043$, partial $\eta^2 = 0.133$). When considering those trials in which we detected a microsaccade, we found that performance was significantly better (smaller errors) in trials in which the first detected microsaccade in our window was toward ($M = 13.65$) vs. away from ($M = 14.72$) the memorised location of the to-be-attended memory item ($t(22) = -2.96$, $P_{Bonferroni} = 0.022$, $d = -0.617$). No significant differences were observed between toward and no-microsaccade trials ($t(22) = -1.41$, $P_{Bonferroni} = 0.52$, $d = -0.293$), or between away and no-microsaccade trials ($t(22) = 1.04$, $P_{Bonferroni} = 0.931$, $d = 0.217$).

We did not find any significant differences when considering the time of response initiation after the attentional cue (Fig. 2b; $F(2, 44) = 0.659$, $P = 0.522$, partial $\eta^2 = 0.029$; all Bonferroni-corrected post hoc $t$-tests $Ps = 1$). This suggests that any potential differences in neural activity after the selection cue between toward, away, and no-microsaccade trial classes are unlikely to be attributed to overall differences in the time between cue-onset and report onset.

These data provide the appropriate context for interpreting the neural modulations that are the central focus of our article. They show that (1) performance is comparable in trials with and without microsaccades (suggesting that performance on our task may not critically depend on the presence/absence of micro-saccades in the attentional selection period of interest), but also

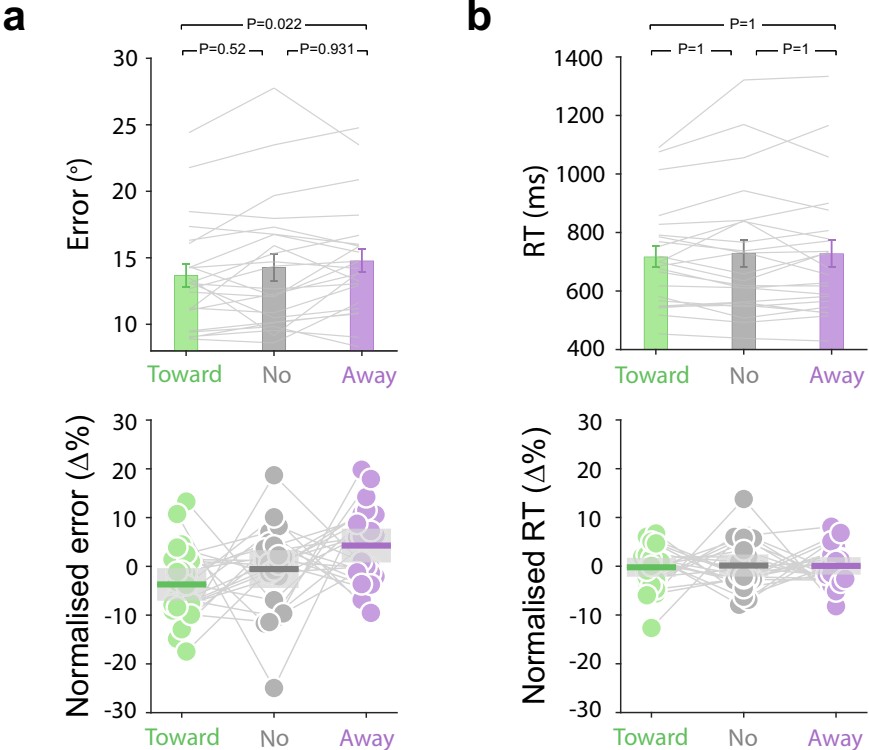

**Fig. 2 Behavioural performance as a function of microsaccade direction and presence. a** Mean reproduction errors (the absolute deviation between the target orientation and the reported orientation in degrees) and **b** mean reaction times (the time between cue onset and report onset) in trials in which the first-detected microsaccade in the defined 200–600 ms selection period was toward (green) or away (purple) from the memorised location of the cued memory item, or in which no microsaccade was detected (grey). Top panels show raw performance with grey lines indicating individual-participant data, while the bottom panels show normalised performance (percent change from mean) together with individual data points. Error bars in the top panels indicate ±1SEM. Shadings in the bottom panels indicate 95% confidence interval. Both the SEM and confidence interval are calculated across participants ($n = 23$). The statistical tests used in the figure were two-sided paired samples $t$-test. The reported $P$ values are all Bonferroni corrected. Source data are provided as a Source Data file.

that (2) when a microsaccade *is* detected, its direction is indicative of the successful allocation of attention, as reflected in performance. As we detail below, our analysis of 8–12 Hz alpha activity converged on a similar pattern whereby microsaccades are correlated with, but not necessary for, spatial attentional modulation of alpha activity.

**Alpha modulation by covert spatial attention is similar in trials with or without attention-directed microsaccades.** Having established the relation between microsaccades and performance, the central question was whether directional biases in microsaccades were necessary for observing spatial modulation of 8–12 Hz alpha activity—a well-established neural signature of spatial attention in human MEG and EEG measurements[26–30]. This spatially specific signature is constituted by a relative attenuation of power contralateral vs. ipsilateral to the direction of spatial attention, including when directed within the spatial layout of visual working memory[34–40].

Figure 3 shows the time- and frequency-resolved neural modulations relative to the memorised item location in pre-defined posterior (visual) electrode clusters centred around electrodes PO7 (left) and PO8 (right). Trials containing a toward microsaccade (Fig. 3a) showed a clear lateralised modulation of alpha activity (two-sided cluster-based permutation test: $P < 0.001$) with a posterior topography. If attention-directed microsaccades are a prerequisite for this neural signature of spatial attention to occur, then we would expect to see an absence (or at least a marked reduction) of such modulation in the trials

in which we did not observe any discernible microsaccade in the selection period of interest. In stark contrast, no-microsaccade trials showed a clear spatial modulation of alpha activity (Fig. 3b; two-sided cluster-based permutation test: $P < 0.001$) equivalent to the lateralised modulation in the toward-microsaccade trials. This was true even when we extended our window and only considered no-microsaccade trials in which no microsaccade was detected anywhere from 0 to 600 ms relative to cue onset (Supplementary Fig. 5)—thereby also excluding trials in which a microsaccade may have occurred very early after the cue. When we overlaid the alpha-lateralisation time courses in the toward and the no-microsaccade trials (Fig. 3d) we observed an almost perfect overlap (no clusters found, all uncorrected $P$s > 0.14).

**Microsaccadic and neural signatures of covert spatial attention are correlated.** The above results make clear that microsaccades are not necessary for neural signatures of covert spatial attention (here, lateralisation of EEG alpha activity) to occur. This does not imply, however, that microsaccadic and neural signatures of covert spatial attention do not co-vary. In fact, when only considering trials in which we did detect a microsaccade, we found two sources of evidence that alpha and microsaccadic signatures (when present) are correlated.

First, similar to our behaviour results, when only considering trials with microsaccades, the direction of the first-detected microsaccade had an impact on the observed neural signature of spatial attention. As also shown in Fig. 3, when the first-detected microsaccade was observed in the away direction, the alpha

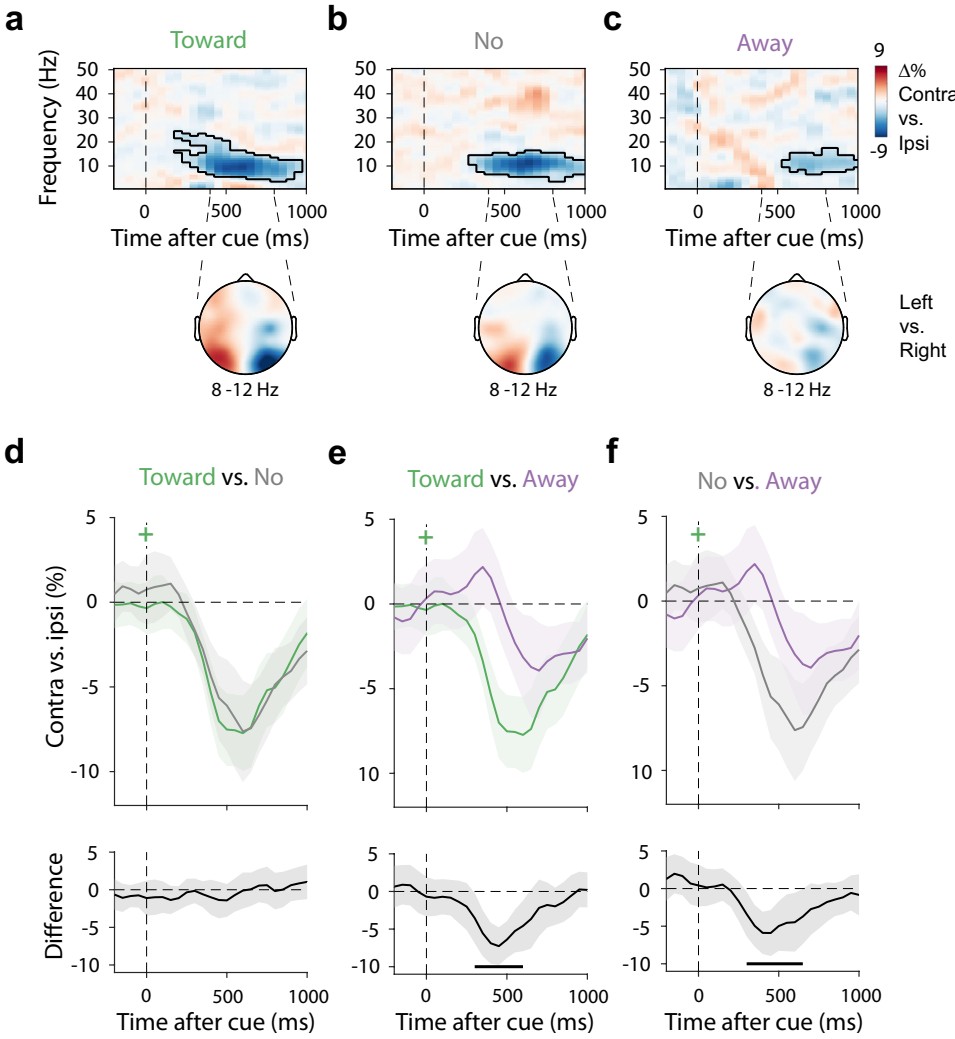

**Fig. 3 Attention-directed microsaccades are correlated with, but not necessary for EEG-alpha modulation by covert spatial attention. a–c** Neural lateralisation in visual electrodes relative to the memorised location of the memory item after the selection cue (top) together with the associated topographical map of the difference in 8–12 Hz alpha power in the 400–800 ms interval between trials in which to-be-reported memory item was left vs. right at encoding (bottom). This is shown separately for trials with a **a** toward microsaccade, **b** No microsaccade, or **c** an away microsaccade in the predefined 200–600 ms selection period (Fig. 1d). Outlines indicate significant time-frequency clusters (two-sided cluster-based permutation test). **d–f** Overlays and comparisons of averaged 8–12 Hz alpha lateralisation. Black horizontal lines indicate significant temporal clusters (two-sided cluster-based permutation test). Time courses show mean values, with shading indicating 95% CI (calculated across 23 participants).

modulation (Fig. 3c; cluster $P = 0.002$) was attenuated compared to both the toward (Fig. 3e; cluster $P < 0.001$) and the no-microsaccade (Fig. 3f; cluster $P = 0.001$) conditions. These results suggest that when microsaccades occur in the "wrong" (away) direction (putatively reflecting attention being initially allocated to the wrong item), the alpha modulation is also relatively reduced. This is in line with a correspondence between the two types of signatures (i.e., each reflecting the allocation of covert spatial attention, even if not directly causing each other).

Second, if these two signatures are related in terms of the underlying function they reflect, they may also co-vary in their timings after the cue (related to spontaneous variability in the timing of attentional selection across trials). To test this, we sorted all trials with a toward microsaccade based on micro-saccade latency. That is, whether the onset of the first-identified toward microsaccade was earlier or later than the median latency of all detected toward microsaccades for that participant. Figure 4a shows the group-level distribution of microsaccade onset latencies in "early" ($M = 324$ ms) and "late" ($M = 473$ ms) trial classes

(note that the overlap in this group-level visualisation reflects the fact that the median latency is different in different participants; Supplementary Fig. 4). While we sorted our trials based purely on microsaccade latency, we found corresponding differences in the timing of alpha activity (Fig. 4b) (as well as in the onset times of the behavioural report after the memory cue; Supplementary Fig. 5). Alpha lateralisation occurred earlier in early-microsaccade trials (Fig. 4b), giving rise to a significant difference in alpha lateralisation between early- and late-microsaccade trials (two-sided cluster-based permutation test: $P = 0.04$). When directly quantifying the latency of the alpha modulation, we found that this modulation reached its half-peak latency 100 ms earlier in early (half peak at 355 ms) vs. late (half peak at 455 ms) toward-microsaccade trials (dashed vertical lines in Fig. 4b). This latency difference was highly significant (permutation-based latency analysis: $P = 0.006$).

These results thus provide converging evidence for functional co-variation— even if not an obligatory link—between the neural and microsaccadic signatures of covert spatial attention. In

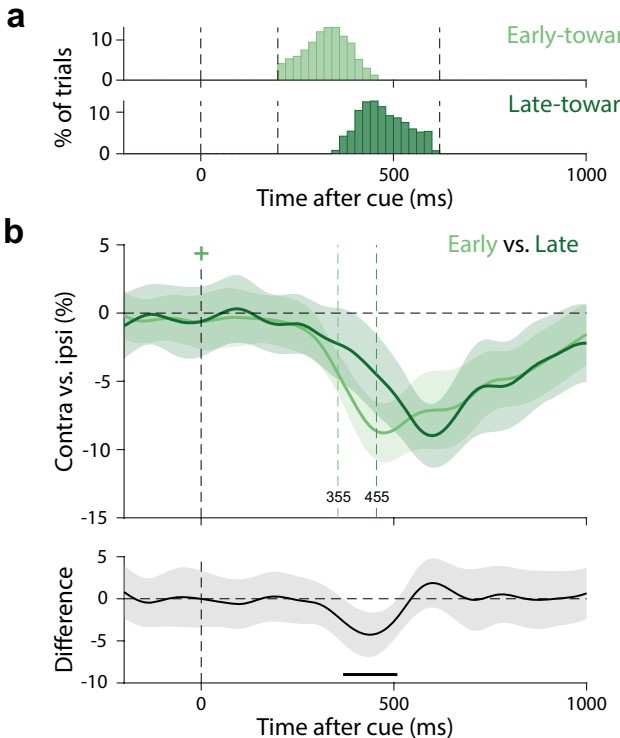

**Fig. 4 Microsaccadic and neural modulations by internally directed spatial attention co-vary in their timings after the selection cue. a** Group-level distributions of microsaccade onset latency in trials with early (top) and late (bottom) microsaccades. Early and late were defined based on a median split that was performed separately for each participant (resulting in some overlap in this group-level visualization). **b** Overlay and comparison of the spatial alpha modulation in toward-microsaccade trials in which the first toward microsaccade was detected relatively early or late. Black horizontal line indicates the significant temporal cluster (two-sided cluster-based permutation test). Time courses show mean values, with shading indicating 95% CI (calculated across 23 participants).

addition, these comparisons of trials with toward vs. away and with early vs. late microsaccades replicate and extend earlier work on attentional signals in invasive recordings in non-human primates[21] to the attentional modulation of scalp-EEG alpha activity in humans.

**Replication of our main findings when using another microsaccade-detection method.** We replicated our main findings using an alternative, more established method for detecting microsaccades (as described in ref. [20]) that considered both horizontal and vertical displacements in gaze in both eyes. This further allowed us to validate our method and to reinforce its suitability for our aims. We report and discuss these data in Supplementary Fig. 7 and its associated supplementary discussion.

**Discussion**
To examine the link between spatial modulations of neural activity and of microsaccades by covert attention, we examined simultaneously collected EEG and eye-tracking data when attention was directed internally within the spatial layout of visual working memory. We found that both the direction and the timing of microsaccadic and EEG-alpha modulations of spatial attention are correlated. At the same time, our results showed that trials with no discernable attention-driven microsaccades nevertheless showed preserved alpha modulation by covert spatial

attention. This shows that the presence of biased microsaccades is not a prerequisite for neural modulation by covert spatial attention to occur.

Our data build on a vast literature linking covert spatial attention to the oculomotor system[5–20]. At the same time, our data show that microsaccades—reflecting output of the oculomotor system—are not necessary for neural EEG-alpha modulation by covert attention to occur. This, however, does not imply that the oculomotor system is not involved in covert selective attention in these trials without an overt microsaccade. Our data suggest that covert attention increases the propensity for microsaccades to occur in the direction of the (memorised) location of attended objects. This directional biasing of microsaccades, however, will only occur when activity in the oculomotor system reaches the threshold for outputting a microsaccade. At times at which activity in the oculomotor system is not close to outputting a saccade, the same oculomotor system may nonetheless contribute to shifting attention without concomitantly yielding an overt reflection in the form of a microsaccade[8,41,42]. Thus, directionally biased microsaccades and neural modulation may reflect two separate manifestations of covert selective attention operations that may each be mediated (at least in part) by the same oculomotor circuitry. When crossing the threshold for an overt microsaccade, the timing and direction of microsaccades functionally co-vary with the timing and direction of EEG-alpha modulation (and other neural signatures such as spike-rate; as in refs. [21,42]). Future work targeting neural activity in and outside the oculomotor system in trials with and without microsaccades is required to substantiate this threshold interpretation.

Our data help reconcile prior behavioural studies that have linked microsaccades to covert spatial attention with contradicting interpretations. On the one hand, studies have argued that microsaccades are tightly correlated with the direction of covert attention[13–20]. At the same time, other studies have pointed out that microsaccades do not always reflect the attention shift, and may not serve as a reliable index of covert attention[43,44]. Our analysis, which also considered neural modulation during the deployment of covert attention, are compatible with both views because they show that microsaccades are correlated with (cf.[13–20]) but at the same time not necessary for (cf.[43,44]) neural modulation by covert attention to occur (see also ref. [42]). Just like our data reveal how neural modulation by covert attention may occur without concurrent microsaccades, complementary studies have suggested that not all microsaccades are necessarily associated with shifts of attention[44,45].

Building on the above prior behavioural work, we specifically focused on the link between microsaccade biases and neural modulation by covert spatial attention. Our analysis was motivated by a recent study in non-human primates by Lowet et al.[21] that reported that the modulation of neural activity by covert attention occurs only when paired with microsaccades toward the attended location. In correspondence with Lowet et al., we demonstrate that (like the firing-rate modulation in V4) the spatial modulation of EEG-alpha activity also co-varies with the direction and the timing of microsaccades (in microsaccade-present trials). Critically, however, we also show that microsaccades are not necessary for the spatial modulation of neural activity to occur.

When directly comparing our study and analysis approach to the one in Lowet et al.[21], there are several notable differences. We studied a different species (human vs. non-human primate) and considered a different neural signature of covert attention (EEG-alpha vs. firing-rate modulations). However, if the dependence of neural markers of covert attention on microsaccades is meant to be a general principle, then the aforementioned differences

should not account for the discrepancy in our conclusions. In contrast, some of the experimental methods we introduced may hold the key. First, we explicitly considered 'no-microsaccade' trials (in which no discernable microsaccade was identified in the attentional window of interest) to arrive at our conclusion that neural modulation by spatial attention may not hinge on the concurrent presence of microsaccades. Second, we used a non-spatial colour cue to direct attention. In contrast, Lowet et al. used a cue that itself contained a spatial element (a dotted line directed to the target). Accordingly, the microsaccades studied by Lowet et al. may have at least partly reflected actual looking at the cue itself before subsequently allocating covert attention to the correct (cued) object.

To address how neural signatures are related to microsaccadic signatures of covert spatial attention in humans, we focused on modulation of alpha activity, a canonical neural signature of covert spatial attention in M/EEG measurements[26–30,34]. Despite the well-established links between covert attention and alpha activity on the one hand[27–30], and directional biases of micro-saccades on the other hand[15,20] it had remained unclear how these two signatures are related. Complementary to prior studies that have already linked the Frontal Eye Fields—a key structure in the oculomotor system—to spatial alpha modulation[31–33], we here also link directional microsaccade biases to alpha modulation. At the same time, we show that alpha modulation is not a direct consequence of attention-directed microsaccades. Nevertheless, the fact that alpha modulation functionally co-varies with the timing and direction of microsaccades suggest oculomotor activity may be an important factor to consider in studies linking alpha activity to various cognitive functions (see also ref. [46]).

Our focus here was on whether microsaccades account for established neural modulation by spatial attention. This complements previous work focusing on the neural modulation by a microsaccade itself (e.g.,[23,47]). For example, Dimigen et al.[47] showed how microsaccades may evoke specific ERP components and modulate alpha activity. While such modulations may contribute to our results (e.g., to the observed co-variation in timing), our critical insight comes from the no-microsaccade trials. Showing preserved alpha modulation in these critical trials led to our main conclusion that directionally biased microsaccades are not necessary for spatial modulation of alpha activity to occur.

We studied the link between microsaccades and EEG-alpha modulation in a task in which attention was directed internally within the spatial layout of working memory. Though not as mainstream as the study of externally directed selective attention, this brings relevant advantages as a model system for studying covert attention and its microsaccadic and neural signatures. First, any attention-driven modulation observed in neural activity cannot be due to retinal effects associated with the attended stimulus moving closer to the fovea, as the object of attention exists in the mind, not the world. Second, unlike when covertly selecting peripheral visual objects, when selecting internal visual representations in working memory there is no incentive for overt gaze behaviour as there is no external object to look at. Therefore, the robust microsaccade biases we observed are a relatively 'pure' reflection of oculomotor engagement. Third, while the set-up may appear distinct from the typical 'covert attention' set-up with external objects, similar neural (i.e. alpha) and microsaccadic modulations can be studied as those studies during externally directed covert spatial attention[15,20,26–28,30,34]. Finally, recent work has shown how such internally directed attention can also be studied in the non-human primate[3], which opens important novel opportunities[48]. Nevertheless, a relevant question for future research is whether/how the link between microsaccades and neural signatures of covert attention may depend on the internal vs. external locus of attention.

Finally, our analysis and interpretations are contingent on how well we were able to detect the presence/absence of micro-saccades, as well as the suitability of our microsaccade window of interest. Admittedly, it is notoriously hard to detect all saccades and miss none, as well as to set the perimeters of the temporal window. Nevertheless, we adopted three critical steps to increase confidence in our classification of the critical 'no-microsaccade' condition. First, we only considered trials with a usable eye trace when making this classification. Second, to avoid missing any potential microsaccade, we deliberately set a low threshold (three times the median gaze velocity in the sampled data) for marking microsaccades. Finally, we chose to consider microsaccades in the relatively large window from 200 to 600 ms after cue onset. Nevertheless, we cannot be absolutely sure that we did not mis-classify the occasional minute microsaccade as a no-microsaccade trial, or that some earlier/later microsaccades may have also contributed to the observed neural modulation in no-microsaccade trials. However, even if we missed some true microsaccades in this condition, one would still expect that the no-microsaccade condition would have relatively fewer micro-saccades compared to the toward-microsaccade condition. Hence, if microsaccades would be strictly necessary for spatial modula-tion of EEG-alpha activity, one would expect to observe at least some reduction in this alpha modulation in no-microsaccade trials. Yet, we found that the alpha modulations in toward- and no-microsaccade trials were indistinguishable.

In summary, our data show that microsaccadic and neural markers of covert spatial attention are functionally correlated, but not obligatorily linked. They further demonstrate how selective attention within working memory provides a good model system for studying the microsaccadic and neural signatures of covert attention, and they link 20 years of research that have independently related covert attention with spatial modulations of microsaccades and EEG-alpha activity.

## Methods

The current study was based on a new analysis of the data from an experiment in which we simultaneously recorded electroencephalography (EEG) and eye-tracking data. These two kinds of data have been separately reported in two prior articles addressing distinct questions. Specifically, the EEG data were used to investigate the electrophysiological signature of working-memory guided behaviour[35] while the eye-tracking data were used to demonstrate the presence of microsaccadic signatures of attentional selection in working memory[17]. Here, we analysed these data together to investigate how spatial modulations of microsaccades and posterior 8–12 Hz alpha activity are related.

**Participants**. Twenty-five healthy human volunteers participated in the experiment (age 19-36; 11 male, 2 left handed). The sample size was set based on the planned EEG study, in which the sample size was chosen based on previous publications from our lab that had similar designs and focused on similar neural signatures (for example see ref. [40]). As reported in our previous eye-tracking article using the same dataset (Experiment 1 in ref. [17]), two participants were excluded due to the poor quality of their eye-tracking data. The experimental procedures were reviewed and approved by the Central University Research Ethics Committee of the University of Oxford. All participants provided written informed consent prior to participation and were reimbursed 15 GBP per hour as a compensation for their time.

**Stimuli and procedure**. We employed a covert spatial attention task requiring participants to select and report one of two visual item representations maintained in working memory. Each trial began with an encoding display that contained two to-be-memorised bars (with different colours and orientations) for 250 ms, followed by a memory delay in which only a central fixation cross remained on the screen for a variable delay time that was randomly drawn between 2000 and 2500 ms. Then, the central fixation cross changed colour acting as the attentional selection cue. Participants were asked to report the orientation of the colour-matching memory item by adjusting a central reporting dial using a manual response of either the left (for left-tilted memory items) or right (for right-tilted memory items) hand. The experiment was programmed in Presentation (version 18.3, Neurobehavioral Systems Inc., Berkeley, CA). During the experiment, participants sat in front of a monitor (with a 100-Hz refresh rate) at a viewing distance of ~95 cm with their head resting on a chin rest.

In the encoding display, the two bars (size: 5.7° × 0.8°) were centred at 5.7° to the left and right of fixation. In each trial, the bar were randomly allocated two distinct colours which were drawn from a set of four: blue (RGB: 21, 165, 234), orange (RGB: 234, 74, 21), green (RGB: 133, 194, 18) and purple (RGB: 197, 21, 234), and two distinct orientations (one left tilted and one right tilted) between ±20° to ±70° (avoiding 20° from horizontal and vertical). Bars in the encoding display were equally likely to be cued for report, and the left/right location of the cued memory item was randomized across trials.

The dial-up report (orientation reproduction) was performed by pressing and holding down 1 of 2 keys on the keyboard (the '\' key to rotate the dial leftward and the '/' key to rotate the dial rightward) and the response was terminated at key release. The reporting dial always appeared centrally at response initiation. Feedback was given immediately to participants after response termination by changing the colour of the fixation cross to green for 200 ms for reproduction errors less than 20°, and to red otherwise.

After practising the task for 5–10 min, participants completed two consecutive sessions (1 h for each). Each contained 10 blocks of 60 trials. In each block, the location of cued item was set to have equal trial numbers. Thus, in total, each participant completed 1200 trials including 600 trials in which the central colour cue directed attention to the memory item that had occupied the left location at encoding ("left memory item") and 600 trials in which the cue directed attention to the right memory item.

After every task block, participants completed a custom gaze calibration module in which they were required to look at a small white calibration point that sequentially changed position every 1–1.5 s. The point visited 7 positions in a randomized order: left-top, left-middle, left-bottom, right-top, right-middle, right-bottom and the centre of the screen. Calibration positions other than at the centre were positioned at 5.7° away from the centre (in both the horizontal and the vertical axes), with the left and right locations corresponding to the centre locations of the bars used in the memory tasks.

**Analysis of behavioural data**. Reproduction errors were defined as the absolute difference (in degrees) between the reported orientation and the actual orientation of the cued item. Response time was defined as the time from cue onset to report initiation. Then, a two-step trimming procedure was applied. First, trials with RTs longer than 3000 ms were excluded. Next, data were trimmed based on a cut off value of 2.5 SD from the mean per participant. Analysis of the behavioural-performance data only considered the trimmed data.

**Eye-tracking acquisition and pre-processing**. The eye tracker (EyeLink 1000, SR Research) was positioned ~15 cm in front of the monitor (~80 cm away from the eyes) on a table. Horizontal and vertical gaze positions were continuously recorded for both eyes at a sampling rate of 1000 Hz. Before recording, the eye tracker was calibrated through the built-in calibration and validation protocols from the Eye-Link software.

Offline, eye-tracking data were converted from the native.edf to the.asc format, and eye-tracking data were read into MATLAB through Fieldtrip. Data from the left and right eyes were averaged to obtain a single horizontal and a single vertical gaze-position channel. Blinks were marked by detecting NaN clusters in the eye-tracking data. Then the data during 100 ms before and after detected NaN clusters were rewritten as NaN to eliminate residual blink artefacts. Finally, data were epoched around cue onset.

To increase interpretability, gaze position was normalized using the data from the custom calibration modules. For each participant, first, a reference coordinate was built by using the median gaze position values at each of our seven calibration positions in the window of 500–1000 ms after calibration point onset (allowing participants 500 ms to reach this point after it changing position). Then the raw eye-position data were scaled by this coordinate and converted into ±100% (corresponding to ±5.7° which matched the eccentricity of the centre of the memory items in the task). Thus, after normalization, ±100% in the horizontal gaze signal corresponded to looking at the centre of the original location of either memory item.

**Detecting gaze shifts and sorting trials based on gaze-shift type**. Because the memory items were always horizontally arranged—with one item to the left and the other to the right of fixation—our analysis focused exclusively on the horizontal channel of the normalized eye data. To identify horizontal gaze shifts (see also Supplementary Fig. 1 for a visual depiction of the steps described below), we first quantified the velocity of gaze by using the absolute value of the temporal derivative of gaze position. We then smoothed the velocity using a Gaussian-weighted moving average filter with a 7-ms sliding window (using the built-in function "smoothdata" in MATLAB). Then, we defined the samples in which velocity exceeded a trial-based threshold of three times the median velocity. We deliberately set this low threshold to increase certainty in our subsequent classification of 'no-microsaccade' trials. Furthermore, to avoid counting the same eye movement multiple times, we imposed a minimum delay of 100 ms between successive gaze shifts. Shift magnitude was calculated by estimating the difference in gaze position before (−50–0 ms) vs. after (50–100 ms) the identified shift sample associated with the initial threshold crossing. Depending on the side of the cued item and its

relation to shift direction, we labelled gaze shifts as "toward" or "away" (with toward and away from being defined exclusively with regard to the memorised item location).

We used "gaze shift" to refer to any detected displacement in gaze, but "microsaccade" when referring to the observed directional bias in gaze, given our empirical result that the directional gaze bias of interest is driven almost exclusively by gaze shifts below 1 degree (Fig. 1c). Also note how the outcomes of our gaze-detection procure is binary (shift on/off), but that we also retain shift-size. Specifically, at threshold-crossings, we mark not only whether there is a shift ("on"), but also what size it has (magnitude). This allowed us to create plots of gaze-shift frequency as a function of gaze-shift size, as in Fig. 1c.

We only considered gaze shifts with an estimated magnitude of at least 1% (0.057°). In addition, we marked all trials in which the eye time series contained a NaN (typically caused by a blink or temporary loss of the eyes during the recording) in the period of 0–600 ms post-cue as unusable. These two sets of trials—that contained a NaN anywhere in the window of interest, or in which we did detect a gaze shift but where the shift was <1%—were dismissed prior to our microsaccade trial classification and were not considered further.

We quantified time courses of gaze-shift rates (number of shifts per second) using a sliding time window of 50 ms in steps of 1 ms. Shift-rate time series were additionally decomposed into a time-magnitude representation (quantified as the number of shifts per second at a given shift magnitude at a given time) separately for toward and away shifts. For magnitude sorting, successive magnitude bins of 5% (in normalized space), ranging from 1–110% in steps of 1% were used.

After getting the usable eye traces, we first classified the trial based on whether any discernible gaze shifts were detected in the 200–600 ms post-cue period. If one ($64 \pm 3$ % of trials with usable eye traces) or more ($16 \pm 2$%) gaze shifts were present in this period, we sorted the trial based on the direction of the first-detected gaze shift (into "toward" and "away" trials relative to the memorised location of the cued item). Alternatively, if no gaze shift ($20 \pm 4$%) was detected in this period, the trial was defined as a "no-microsaccade" trial.

In our custom microsaccade-detection approach, we focused on detecting gaze shifts in the horizontal axis because our question focused on gaze-shifts that could potentially account for spatial modulation of neural alpha activity that (in our task) was defined exclusively along the horizontal (left/right) axis. To ensure our results were not critically dependent on this decision, we also evaluated our key results when using the more-established microsaccade-detection method by Engert & Kliegl[20] that considered both horizontal and vertical displacements in gaze (as well as considering both eyes). This replicated our main results, while also providing validation of our method, which we had titrated to our specific purpose (see Supplementary Fig. 7 and associated supplementary discussion).

**EEG acquisition and basic processing**. With the sample rate of 1000 Hz, EEG data were acquired from 61 electrodes through Synamps amplifiers and Neuroscan acquisition software (Compumedics). The electrodes were distributed across the scalp according to the international 10–10 system. To help monitor (and later remove) ocular artefacts in the EEG, the vertical electrooculography (EOG) was recorded from electrodes located above and below the left eye, and the horizontal EOG was recorded from electrodes lateral to the external canthi. During acquisition, data were referenced to the left mastoid and filtered between 0.1 and 200 Hz. Later, data were re-referenced to the mean of the right and left mastoids offline.

Data were analysed in MATLAB through a combination of Fieldtrip and custom code. Continuous EEG was first epoched from −1000 to +2000 ms relative to cue onset. We next used independent component analysis (ICA), as implemented in the field trip, to clean the data. Relying on the correlation with EOG signal, we removed ICA components that captured eye blinks, and eye-movements. Then, we visually inspected and removed trials with exceptionally high variance by using "ft_rejectvisual.m" function in Fieldtrip with the 'summary' method.

After cleaning both the EEG and the eye-tracking data, we had $760 \pm 18$ trials ($63 \pm 1.5$ % of all trials) left for our main (combined EEG and eye-tracker) analysis. Of these, we had $335 \pm 21$ (for "toward"), $166 \pm 9$ (for "away"), and $258 \pm 23$ (for "no") trials left for the eye-tracking dependent EEG analysis.

**Electrode and frequency-band selection**. To increase the spatial resolution, we first performed a surface Laplacian transform. Next, to investigate the lateralisation of neural activity relative to the memorised left/right location of the cued memory item (putatively where visual attention was allocated to), we focused our analysis on pre-defined posterior (visual) electrode clusters centred on PO7 on the left (O1, PO3, PO7, P1, P3, P5, P7) and centred on PO8 on the right (O2, PO4, PO8, P2, P4, P6, P8).

For analysis of alpha lateralization, we first decomposed the epoched EEG time series into a time-frequency representation using a short-time Fourier transform of Hanning-tapered data as implemented in Fieldtrip[49]. A 300-ms sliding time window was used to estimate the spectral power between 1 and 50 Hz (in steps of 1 Hz) that was advanced over the data in steps of 50 ms. When comparing the timing of 8–12 Hz alpha power lateralisation (i.e., comparing the early and late microsaccade conditions), we ran this analysis again, this time maintaining a higher temporal resolution (using steps of 1 ms instead). To zoom in on lateralized modulation relative to the memorised location of the cued item, we expressed the

lateralisation of spectral power as normalised contrasts (i.e., ((contra-ipsi)/(contra +ipsi)) × 100) and averaged these contrasts across the left and right electrodes. To extract time courses of lateralization in the alpha band, we averaged the contrasts across the predefined alpha band between 8 and 12 Hz. To obtain topographical maps of lateralization, we contrasted alpha power for left and right attention trials (i.e., ((left-right)/(left+right)) × 100) for all electrodes.

**Statistical analysis**. Statistical evaluation for the behavioural data was achieved by performing a one-way repeated-measures ANOVA (across the conditions toward, no, and away) complemented with Bonferroni-corrected post hoc t-tests. Cohen's d was used as a measure of effect size.

Statistical evaluation for time-series data used a cluster-based permutation approach[50], which is ideally suited to evaluate the reliability of neural patterns at multiple neighbouring data points while circumventing the multiple-comparisons problem. We used this approach here to estimate the neural patterns across time and (for the spectral analysis) frequency. We first created a permutation distribution of the largest clusters that were found after random permutations of the trial-average data at the participant level. Then, the proportion of permutations for which the size of the largest cluster after permutation is larger than the size of the observed cluster in the original (non-permuted) data gives the P value for the compared cluster. We used 10,000 permutations to create the permutation distribution and identified the cluster with Fieldtrip's default cluster settings (grouping adjacent same-signed data points that were significant in a mass univariate t-test at a two-sided alpha level of 0.05, and defining cluster size as the sum of all t values in a cluster).

To compare the timing of alpha-power lateralisation, we additionally used a custom permutation approach. For this analysis, we only considered trials with a toward microsaccade and split the trials based on the median latency of the first toward microsaccade. Then, we calculated the timing of the alpha modulation in early and late microsaccade trials by marking the timing when alpha lateralisation first reached half of its peak modulation. To test this latency difference statistically, we then used a permutation approach. We permuted the condition label (i.e., early and late microsaccade) at the subject level and again calculated the latency difference in the permuted data as above. Then, we calculated a P value under the permutation distribution of observed latency differences as the proportion of permutations for which the timing difference was larger than the observed difference in the latency of the alpha lateralisation in the original (non-permuted) data. As before, we used 10,000 permutations.

**Reporting summary**. Further information on research design is available in the Nature Research Reporting Summary linked to this article.

## Data availability
All data are publicly available through the Dryad Digital Repository. The eye data used in this study have been deposited under accession code (https://doi.org/10.5061/dryad. m99r286[51]; Experiment 1). The corresponding EEG data used in this study have been deposited under accession code(https://doi.org/10.5061/dryad.sk8rb66[52]). Source data are provided with this paper.

## Code availability
Relevant code associated with the here-presented analyses are available at: https://github.com/BaiweiLiu/VWMMS_EEG_alpha-covertAttention/releases/tag/v.1[53].

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

## Acknowledgements
This research was supported by an ERC Starting Grant from the European Research Council (MEMTICIPATION, 850636) to F.v.E., a Wellcome Trust Senior Investigator Award (104571/Z/14/Z) and a James S. McDonnell Foundation Understanding Human Cognition Collaborative Award (220020448) to A.C.N, and by the NIHR Oxford Health Biomedical Research Centre. The Wellcome Centre for Integrative Neuroimaging is supported by core funding from the Wellcome Trust (203139/Z/16/Z). The authors also wish to thank Sammi Chekroud for assistance in data collection and Rose Nasrawi for valuable input on the manuscript. This research was funded in part by the Wellcome Trust [Grant numbers 104571/Z/14/Z, 203139/Z/16/Z]. For the purpose of open access, the author has applied a CC BY public copyright licence to any Author Accepted Manuscript version arising from this submission.

## Author contributions
F.v.E. programmed the experiments and acquired the data. B.L., A.C.N. and F.v.E. conceived the analysis. B.L. and F.v.E. analysed the data. B.L., A.C.N. and F.v.E. interpreted the data. B.L., A.C.N. and F.v.E. drafted and revised the manuscript.

## Competing interests
The authors declare no competing interests.
