## [Peer Review File · Nature Communications]

Functional but not obligatory link between microsaccades and neural modulation by covert spatial attentionREVIEWER COMMENTS

Reviewer #1 (Remarks to the Author):

This paper shows that attention-related alpha asymmetry does not depend on the occurrence of lateralized microsaccades, despite the tight link found in many previous studies between microsaccades and attention. The findings clearly show that this link is not obligatory; alpha asymmetry was found even in trials where no microsaccade was detected.

The findings are convincing and would contribute to the field. The paper is well-written and concise, and I have enjoyed reading it. I do have a few comments.

1. First, I'm a bit concerned with the use of a new microsaccade detection method that is not validated in this study or compared to other more common methods used in this field. The method is based on velocity threshold, similar to other common detection methods, but the approach is different in some respects. For example, averaging the two eyes is not a common practice in the microsaccade literature, as using binocular information is thought to enhance the SNR. I also should say I found it hard to follow the details of how the detection was done. Specifically, I was confused by the changing terminology between "gaze shifts" and microsaccades. Do they mean the same? I also found it difficult to understand whether the outcome of the procedure is binary (onsets of microsaccades) or continuous (sizes of gaze shifts).

My recommendation is to either replace or validate this method with another method that is more common in the microsaccade literatures (e.g. the method by Engbert et al.). I find this important for two reasons. First, this study is all about detection of microsaccades. I would make the extra effort to convince that everything was done to ensure that the "no microsaccade" trials indeed include no microsaccades. Going for a more common tool (instead or in addition to the one used here) would help towards that aim. Especially since the study by Lowett et al. employed Engbert's method. Second, what makes me more concerned about this detection procedure is that the microsaccade traces shown in Figure 1b do not look like typical saccade traces. These traces do not show the typical post-stimulus inhibition, which we find in 100% of the participants and is typically very robust with the kind of stimulus that was used here (change of color of the fixation cross). Not seeing this inhibition here raises a red flag and I would look deeper into that.

2. Second, I'm not comfortable with the decision to include only horizontal microsaccades in the analysis. Especially because, if I understand correctly, this means that the "no microsaccade" may include vertical microsaccades. This is concerning because with minor calibration shifts, horizontal microsaccades may seem more vertical than they actually are. I would reexamine the data to make sure that the findings hold even when no vertical microsaccades are included in the "no microsaccade" group.

3. Third, I was expecting a discussion of Lambda activity - visual activity evoked by the gaze shift associated with microsaccades. Such visual responses are shown to manifest also as alpha activity (see Dimigen et al., 2009, J Neuro, and a poster by the same group:

[https://www.researchgate.net/profile/Olaf-](https://www.researchgate.net/profile/Olaf-Dimigen/publication/356001733_Microsaccades_EEG_alpha_oscillations_A_close_relationship/links/6189647061f0987720707bcc/Microsaccades-EEG-alpha-oscillations-A-close-relationship.pdf)

[Dimigen/publication/356001733_Microsaccades_EEG_alpha_oscillations_A_close_relationship/links/6189647061f0987720707bcc/Microsaccades-EEG-alpha-oscillations-A-close-relationship.pdf](https://www.researchgate.net/profile/Olaf-Dimigen/publication/356001733_Microsaccades_EEG_alpha_oscillations_A_close_relationship/links/6189647061f0987720707bcc/Microsaccades-EEG-alpha-oscillations-A-close-relationship.pdf)).

Specifically, I wonder whether the "early" "late" findings could be explained as early vs. late visual effects related to the timing of the microsaccades?

I also have a few minor comments:

1. In our study from 2014 (Yuval-Greenberg, Merriam and Heeger), we have found a link between the direction of microsaccades and spontaneous shifts of attention. But, importantly for the purpose of this paper, we claim that although there is a link between attention and microsaccades, not every microsaccade is linked to attention. Together with the present findings, this could offer a complete dissociation: attention can be shifted without microsaccades, microsaccades can be shifted without attention but they are both nevertheless linked in most cases.

2. The sentence that "no study has systematically related these two recognised signatures of covert spatial attention" is inaccurate because of Dimigen's studies cited above.

3. What is the measurement unit of supplementary figure S1 "gaze position"? Looks like pixels, but I would recommend using visual degrees.
4. I would recommend using a sliding window of 50ms and not 100. Or another looser method like the one proposed by Rolfs et al., 2008 (J Vis). The saccade rate trace in Fig. 1b is too smooth and not very informative. Hopefully with less filtering the post-stimulus inhibition would show up.
5. In the list of differences between this study and the study by Lowett (Discussion), I would add the different microsaccade detection procedure.

I would be happy to answer questions if something is unclear.
Congratulations for a very nice work!
Shlomit Yuval-Greenberg

Reviewer #2 (Remarks to the Author):

This is a particularly nice research report in which the relation between attention-related micro-saccades and lateralised EEG alpha activity is investigated. For the first time, two very well documented phenomena are brought together, namely, the tendency of micro-saccades towards spatial locations to which spatial attention is directed and alpha amplitude decrease contralateral to the attended hemifield. This study shows that there is comodulation of these processes but that they are not necessarily coupled. There is a substantial number of trials in which no micro-saccades are made or where micro-saccades were performed away from the attended location. Still, in these trials reliable alpha amplitude lateralisation was obtained. That is an important result showing that these two mechanisms are causally independent from each other.

The manuscript is particularly well written. It is easy to follow, concise and well-structured. Although the general idea is simple the study is innovative and will have strong impact. I like the experimental paradigm which encourages shifting of attention without visual stimulation in the critical time window. The results are very convincing, and the conclusions are sound. This truly is one of the nicest manuscripts I have read in some time.

I only have one very minor suggestion: In fig 1 it is not really clear why individual data points are not shown for raw error and RT. It would be good to get a feel for the distribution similar as shown in the lower panels (on normalised error and RT).

Reviewer #3 (Remarks to the Author):

The authors have used an intriguing behavioral paradigm with EEG recordings in humans to investigate the link between spatial attention and microsaccades. They previously showed that subjects tend to make microsaccades (MS) in the directions of items held in working memory, i.e. not actually present on the screen. Attention to the item in working memory also modulates the laterality of alpha power in the EEG, so they are able to test the links between attention, MS, and a neural marker of attention. They find that alpha power is strongly modulated by the direction of the MS, consistent with physiological data reported in monkeys, but they also find modulation of alpha power when there is attention directed to the internal stimulus, but no MS. I have just two questions.

1. Although they find attentional modulation even during analysis windows with no MS, an important question is whether a MS was made just prior to the analysis window. It has been shown that an early MS can influence neural signals in a later window. The authors should examine an earlier window of at least 200 ms, to see if a MS was made during that time on the "no MS" trials. Figure 2 gives the impression that there are no MS made prior to 200 ms, but my understanding is that MS are shown only during the 200-600 ms analysis window. A MS might even have been made before the attentional

cue. If I am correct, then the figure is actually misleading. Please clarify and correct if needed.

2. As the authors acknowledge, the study relies at least in part on the ability to correctly detect MS. To get some sense of whether they are being correctly detected, it would be useful to see some plots and statistics on overall MS frequency and direction, to see if the MS characteristics are similar to what have been found in other studies.

Dear Reviewers,

We were very pleased with the overall enthusiasm for our article and findings, as well as for each of the constructive comments we received. Having embraced this valuable feedback, we have been able to further strengthen the quality of our article. Specifically, we have now included the outcomes of relevant additional analyses that reinforce our original conclusions and analysis choices, and that further enhance the comprehensiveness of our manuscript.

We are delighted to resubmit our improved article with revisions highlighted in blue. In addition, please also find our detailed point-by-point replies below.

We are (again) grateful for your time in considering our article.

Sincerely,

Baiwei Liu, Kia Nobre, Freek van Ede

Reviewer 1

This paper shows that attention-related alpha asymmetry does not depend on the occurrence of lateralized microsaccades, despite the tight link found in many previous studies between microsaccades and attention. The findings clearly show that this link is not obligatory; alpha asymmetry was found even in trials where no microsaccade was detected.

The findings are convincing and would contribute to the field. The paper is well-written and concise, and I have enjoyed reading it. I do have a few comments.

Thank you for your positive overall evaluation of our manuscript, as well as for each of your constructive and valuable comments for improvement, to which we respond in detail below.

Prompted by your valuable suggestion, we have now also analysed microsaccades using the complementary (and more established) method by Engbert & Kliegl (2003). As you will see below, these additional analyses have reinforced our original conclusions, as well as provided additional confidence in the suitability of the custom microsaccade-detection approach we had applied.

First, I'm a bit concerned with the use of a new microsaccade detection method that is not validated in this study or compared to other more common methods used in this field. The method is based on velocity threshold, similar to other common detection methods, but the approach is different in some respects. For example, averaging the two eyes is not a common practice in the microsaccade literature, as using binocular information is thought to enhance the SNR. I also should say I found it hard to follow the details of how the detection was done. Specifically, I was confused by the changing terminology between "gaze shifts" and microsaccades. Do they mean the same? I also found it difficult to understand whether the outcome of the procedure is binary (onsets of microsaccades) or continuous (sizes of gaze shifts).

My recommendation is to either replace or validate this method with another method that is more common in the microsaccade literatures (e.g. the method by Engbert et al.). I find this important for two reasons. First, this study is all about detection of microsaccades. I would make the extra effort to convince that everything was done to ensure that the "no microsaccade" trials indeed include no microsaccades. Going for a more common tool (instead or in addition to the one used here) would help towards that aim. Especially since the study by Lowett et al. employed Engbert's method. Second, what makes me more concerned about this detection procedure is that the microsaccade traces shown in Figure 1b do not look like typical saccade traces. These traces do not show the typical post-stimulus inhibition, which we find in 100% of the participants and is typically very robust with the kind of stimulus that was used here (change of color of the fixation cross). Not seeing this inhibition here raises a red flag and I would look deeper into that.

Thank you for raising this important point. We fully agree that it is important to make sure our conclusions are not contingent on our specific method of microsaccade detection. We therefore followed the reviewer's suggestion and additionally evaluated our results when relying on the more established microsaccade-detection method by Engbert & Kliegl (2003). At the same time, this allowed us to validate the method we had developed, which was titrated to our purpose, and to reinforce its suitability for our aims. As we describe below, this replicated our central results, thereby reinforcing our original conclusions. Because we believe this is an important addition to our manuscript, we have added these new results to our manuscript. Specifically, we have added the following key elements to our manuscript:

Page 7 (Results):

Replication of our main findings when using another microsaccade-detection method

We replicated our main findings using an alternative, more established method for detecting microsaccades (as described in Engbert & Kliegl, 2003) that considered both horizontal and

vertical displacements in gaze in both eyes. This further allowed us to validate our method and to reinforce its suitability for our aims. We report and discuss these data in **Supplementary Figure 7** and its associated supplementary discussion.

Page 24/25 (Supplementary Material):

Supplementary Figure 7. Replication of our main results using an alternative microsaccade-detection method. *a*) Time courses of gaze-shift rates (number of microsaccades per second) for shifts toward and away from the memorised location of the cued memory item. The left panel shows the results from Engbert & Kliegl’s 2003 method; the right panel shows the results from our method. *b*) The overlap of trials in each condition that sorted by Engbert & Kliegl’s method and our method. The top row represent the trials sorted by Engbert & Kliegl’s method while the bottom row represent the overlap in trial-allocation when relying on our method. *c*) In each condition, the overlays of averaged 8-12 Hz alpha lateralization from the trials that are sorted by Engbert & Kliegl’s method and by our method.

Supplementary results text associated with Supplementary Figure 7

While we titrated our method of microsaccade detection to our current aims (deliberately using a relatively low threshold for detecting, and focusing on horizontal shifts of gaze), we obtained similar results when using another, more established, method by Engbert & Kliegl (2003) instead.

First, we obtained similar temporal profiles of microsaccades and their modulation by internal selective attention (**Supplementary Fig. 7a**).

Second, we converged on strongly overlapping trial classifications when relying on our method compared to the Engbert & Kliegl method. When the Engbert & Kliegl method detected a toward microsaccade in the 200-600 ms post-cue window of interest, so did our method in the vast majority of trials (**Supplementary Fig. 7b**, left). Likewise, when the Engbert & Kliegl method detected an away microsaccade, so did our method (**Supplementary Fig. 7b**, middle). Moreover, when the Engbert & Kliegl method did not detect a microsaccade (in our window of interest), our method occasionally still detected a toward or away microsaccade (**Supplementary Fig. 7b**, right). This reveals that our method was more strict when it comes to allocating the label “no-

microsaccade” to a trial (i.e. our methods seemed more liberal for allocating gaze shifts, and thereby more conservative for determining no-microsaccade trials). This is an important characteristic given our central aim of addressing whether the neural modulation is preserved in these critical no-microsaccade trials.

Finally, irrespective of these slight differences in trial-allocation between methods, alpha modulation in each of the three trial classes was highly comparable between the two microsaccade-detection methods (Supplementary Fig. 7c) – revealing a clear alpha modulation in the no-microsaccade trials in both cases. These data thus help validate our method, and reinforce our central conclusions by showing that our main findings generalise when using a different microsaccade-detection method.

Please also note how this newly added Supplementary Figure 7 shows how both our original and the Engbert & Kliegl method do reveal a clear initial oculomotor inhibition following the cue. This was not apparent from our original figure (Fig. 1b) because in that visualisation, the same time courses were relatively stretched out across time. The same data is visualised in Supplementary Fig. 7a (see above), where it is considerably more compressed in time. Reassuringly, this reveals the expected pattern of oculomotor inhibition. Moreover, note how this also shows the comparable nature of this inhibition in our data, when analysed with the two distinct methods.

In addition, we have now also added further clarification to our Methods for why we had developed and applied our (related) microsaccade-detection method that was titrated to our purposes:

Page 13 (Methods): *In our custom microsaccade-detection approach, we focused on detecting gaze shifts in the horizontal axis because our question focused on gaze-shifts that could potentially account for spatial modulation of neural alpha activity that (in our task) was defined exclusively along the horizontal (left/right) axis. To ensure our results were not critically dependent on this decision, we also evaluated our key results when using the more-established microsaccade-detection method by Engert & Kliegl (2003) that considered both horizontal and vertical displacements in gaze (as well as considering both eyes). This replicated our main results, while also providing validation of our method, which we had titrated to our specific purpose (see Supplementary Fig. 7 and associated supplementary discussion).*

With regard to terminology, we used “gaze shift” to refer to any detected displacement in gaze, but “microsaccade” when referring to the observed directional bias in gaze, given our empirical result that the directional gaze bias of interest is driven almost exclusively by gaze shifts below 1 degree (see Figure 1c, also depicted below for convenience). In addition, note how the outcomes of our analysis procedure is both binary (shift on/off) and continuous. At threshold-crossings, we mark not only whether there is a shift (“on”), but also what size it has (magnitude). This allowed us to create plots like the key plot in Figure 1c:

Figure 1c.

We have now also clarified these points in our Methods section:

Page 13 (Methods): *We used “gaze shift” to refer to any detected displacement in gaze, but “microsaccade” when referring to the observed directional bias in gaze, given our empirical result that the directional gaze bias of interest is driven almost exclusively by gaze shifts below*

1 degree (see Fig. 1c). Also note how the outcomes of our gaze-detection procedure is binary (shift on/off), but that we also retain shift-size. Specifically, at threshold-crossings, we mark not only whether there is a shift (“on”), but also what size it has (magnitude). This allowed us to create plots of gaze-shift frequency as a function of gaze-shift size, as in Figure 1c.

Second, I'm not comfortable with the decision to include only horizontal microsaccades in the analysis. Especially because, if I understand correctly, this means that the “no microsaccade” may include vertical microsaccades. This is concerning because with minor calibration shifts, horizontal microsaccades may seem more vertical than they actually are. I would reexamine the data to make sure that the findings hold even when no vertical microsaccades are included in the “no microsaccade” group.

Thank you for raising this point. As we clarify immediately above (as part of our response to point 1) – and now also make clear in our Methods section – we deliberately focused on horizontal gaze shifts as we wanted to address to what extent horizontal biases in microsaccades could account for alpha lateralisation which, in our task, was exclusively defined in the horizontal (left/right) axis.

Nevertheless, we agree that it is conceivable that we missed relevant microsaccades by doing so. To take away this potential concern, please note how our newly added analyses (Supplementary Fig. 7) replicated our main findings when using the Engbert & Kliegl method instead – for which both horizontal and vertical gaze shifts were considered. As already made clear above, we now also point this out at the relevant instance in our Methods as well as our Results. For example:

Page 7 (Results): We replicated our main findings using an alternative, more established method for detecting microsaccades (as described in Engbert & Kliegl, 2003) that considered both horizontal and vertical displacements in gaze in both eyes.

Third, I was expecting a discussion of Lambda activity - visual activity evoked by the gaze shift associated with microsaccades. Such visual responses are shown to manifest also as alpha activity (see Dimigen et al., 2009, J Neuro, and a poster by the same group. Specifically, I wonder whether the “early” “late” findings could be explained as early vs. late visual effects related to the timing of the microsaccades?

Thank for pointing us to this relevant prior work. Prompted by this comment we have added the following relevant clarification to our discussion, where we also cite this and other relevant work:

Page 9 (Discussion): Our focus here was on whether microsaccades account for established neural modulation by spatial attention. This complements previous work focusing on the neural modulation by a microsaccade itself (e.g., Dimigen et al., 2009; Yuval-Greenberg et al., 2008). For example, Dimigen et al. (2009) showed how microsaccades may evoke specific ERP components and modulate alpha activity. While such modulations may contribute to our results (e.g., to the observed co-variation in timing), our critical insight comes from the no-microsaccade trials. Showing preserved neural modulation in these critical trials led to our main conclusion that directionally biased microsaccades are not necessary for spatial modulation of alpha activity to occur.

few minor comments:

1. In our study from 2014 (Yuval-Greenberg, Merriam and Heeger), we have found a link between the direction of microsaccades and spontaneous shifts of attention. But, importantly for the purpose of this paper, we claim that although there is a link between attention and microsaccades, not every microsaccade is linked to attention. Together with the present findings, this could offer a complete dissociation: attention can be shifted without microsaccades, microsaccades can be shifted without attention but they are both nevertheless linked in most cases.

Thank you for bringing this relevant work to our attention. We now refer to this relevant related finding at the relevant instance in our Discussion:

Page 8 (Discussion): [...] *Just like our data reveal how neural modulation by covert attention may occur without concurrent microsaccades, complementary studies have suggested that not all microsaccades are necessarily associated with shifts of attention (Yuval-Greenberg et al., 2014; Horowitz et al., 2007).*

2. The sentence that “no study has systematically related these two recognised signatures of covert spatial attention” is inaccurate because of Dimigen’s studies cited above.

Thank you, we agree and have now rephrased this sentence to:

Page 2 (Introduction): [...] *little remains known about how these two separate hallmark signatures of covert spatial attention are related.*

Moreover, as described above (in response to major comment 3), we now also explicitly bring this relevant literature forward in our Discussion.

3. What is the measurement unit of supplementary figure S1 “gaze position”? Looks like pixels, but I would recommend using visual degrees.

Thank you for this constructive suggestion. We have now also included the degrees visual angle in our example data segment in Supplementary Figure 1, where we visualize the key steps in our microsaccade-detection approach.

4. I would recommend using a sliding window of 50ms and not 100. Or another looser method like the one proposed by Rolfs et al., 2008 (J Vis). The saccade rate trace in Fig. 1b is too smooth and not very informative. Hopefully with less filtering the post-stimulus inhibition would show up.

Thank you also for this constructive suggestion that we have also followed up on. Employing this new sliding window of 50 ms indeed makes the post-stimulus oculomotor inhibition clearer (Fig. 1b), which can be best appreciated in Supplementary Figure 7a (also depicted in response to major point 1 above) due to its relatively more compressed timescale.

5. In the list of differences between this study and the study by Lowett (Discussion), I would add the different microsaccade detection procedure.

Thank you also for this constructive suggestion. As described and shown above, we now replicated our key findings with the Engbert & Kliegl method (that was also used by Lowett et al.) and have added these new results to our manuscript. Accordingly, this is no longer a relevant difference between our studies.

Reviewer 2

This is a particularly nice research report in which the relation between attention-related micro-saccades and lateralised EEG alpha activity is investigated. For the first time, two very well documented phenomena are brought together, namely, the tendency of micro-saccades towards spatial locations to which spatial attention is directed and alpha amplitude decrease contralateral to the attended hemifield. This study shows that there is comodulation of these processes but that they are not necessarily coupled. There is a substantial number of trials in which no micro-saccades are made or where micro-saccades were performed away from the attended location. Still, in these trials reliable alpha amplitude lateralisation was obtained. That is an important result showing that these two mechanisms are causally independent from each other.

The manuscript is particularly well written. It is easy to follow, concise and well-structured. Although the general idea is simple the study is innovative and will have strong impact. I like the experimental paradigm which encourages shifting of attention without visual stimulation in the critical time window. The results are very convincing, and the conclusions are sound. This truly is one of the nicest manuscripts I have read in some time.

Thank you for your positive overall evaluation of our manuscript and lovely compliment. We respond to your constructive and valuable comment for improvement below.

In fig 1 it is not really clear why individual data points are not shown for raw error and RT. It would be good to get a feel for the distribution similar as shown in the lower panels (on normalised error and RT).

Thank you for this constructive suggestion. We now also added individual data points/lines to the raw error and RT data. We agree that this improves transparency. This now looks as follows:

Figure 2. Behavioural performance as a function of microsaccade direction and presence. (a) Reproduction errors (the absolute deviation between the target orientation and the reported orientation in degrees) and (b) reaction times (the time between cue onset and report onset) in trials in which the first detected microsaccade in the defined 200-600 ms selection period was toward (green) or away (purple) from the memorised location of the cued memory item, or in which no microsaccade was detected (grey). Top panels show raw performance with grey lines indicating individual-participant data, while the bottom panels show normalised performance (percent change from mean) together with individual data points. Error bars in the top panels indicate ± 1 SEM. Shadings in the bottom panels indicate 95% confidence interval. Both the SEM and confidence interval are calculated across participants ($n = 23$).

Reviewer 3

The authors have used an intriguing behavioral paradigm with EEG recordings in humans to investigate the link between spatial attention and microsaccades. They previously showed that subjects tend to make microsaccades (MS) in the directions of items held in working memory, i.e. not actually present on the screen. Attention to the item in working memory also modulates the laterality of alpha power in the EEG, so they are able to test the links between attention, MS, and a neural marker of attention. They find that alpha power is strongly modulated by the direction of the MS, consistent with physiological data reported in monkeys, but they also find modulation of alpha power when there is attention directed to the internal stimulus, but no MS. I have just two questions.

Thank you for your careful reading and for raising these important questions, which we address in detail below.

1. Although they find attentional modulation even during analysis windows with no MS, an important question is whether a MS was made just prior to the analysis window. It has been shown that an early MS can influence neural signals in a later window. The authors should examine an earlier window of at least 200 ms, to see if a MS was made during that time on the "no MS" trials. Figure 2 gives the impression that there are no MS made prior to 200 ms, but my understanding is that MS are shown only during the 200–600 ms analysis window. A MS might even have been made before the attentional cue. If I am correct, then the figure is actually misleading. Please clarify and correct if needed.

Thank you for raising this relevant point, to which we respond in two steps.

As a first step, we clarify that a baseline rate of microsaccades is present before 200 ms, but that the spatially specific modulation of microsaccades onsets at around 200 ms.

Figure 1c may indeed suggest that there are no microsaccades made before 200 ms after the cue. Critically, however, this is because we depicted the *difference* between toward and away microsaccades in Figure 1c. To clarify this important point, we had included the density plots separately for toward and away microsaccades in Supplementary Figure 2, which we also present below for reviewer's convenience:

Supplementary Figure 2. Gaze-shift rates as a function of gaze-shift magnitudes, split by microsaccade direction. Data on the right as in main Figure 1c. The left two panels show these data separately for toward and away gaze shifts (relative to the memorised location of the cued memory item).

As this shows, microsaccades also occur before 200 ms. This is also evident from the time courses in main Figure 1b, which we also depict below for convenience:

Figure 1b.

Thus, microsaccades occur throughout the trial. However, when we specifically consider the *modulation* of microsaccades by spatial attention (toward > away), this effect emerges from around 200 ms after the cue. This was also the reason we started our window of interest at 200 ms.

We hope that the above clarifies the motivation for our analysis choice to focus on microsaccades detected from 200 ms onwards. Nevertheless, we agree that it is important to demonstrate that our results are not critically dependent on this choice of window.

As a second step, we therefore now also demonstrate that our central result (a robust alpha modulation even in the critical ‘no-microsaccade’ trials) does not depend on our original choice of microsaccade window. To this end, we repeated our analysis of alpha activity in the no-microsaccade trials, but this time considered trials in which no microsaccades were detected in the window from 0 (instead of from 200) ms after the cue (thus also not considering trials with microsaccades before 200 ms). Critically, we still found a clear alpha modulation in these no-microsaccade trials, as we depict below. Because we believe these data are a relevant addition to our manuscript, we have included these as follows:

Page 5 (Results): [...] *This was true even when we extended our window and only considered no-microsaccade trials in which no microsaccade was detected anywhere from 0 to 600 ms relative to cue onset (Supplementary Fig. 5) – thereby also excluding trials in which a microsaccade may have occurred very early after the cue.*

Supplementary Figure 5. Trials with no microsaccade in the 0-600 ms post-cue window also show preserved alpha modulation by covert spatial attention. **a**) Neural lateralisation in visual electrodes relative to the memorised location of the memory item after the selection cue (top) together with the associated topographical map of the difference in 8-12 Hz alpha power in the 400-800 ms interval between trials in which to-be-reported memory item was left vs. right at encoding (bottom). Black outline indicates significant cluster (cluster $p < 0.001$). **b**) Averaged 8-12 Hz alpha lateralization. Black horizontal line indicates significant temporal cluster (cluster $p < 0.001$). Shading indicates 95% confidence interval, calculated across participants ($n = 23$).

2. As the authors acknowledge, the study relies at least in part on the ability to correctly detect MS. To get some sense of whether they are being correctly detected, it would be useful to see some plots and statistics on overall MS frequency and direction, to see if the MS characteristics are similar to what have been found in other studies.

Thank you for raising this important point. This point overlaps with major point 1 of reviewer 1. As we elaborated in response to reviewer 1, we have now validated our microsaccade-detection method by replicating our main results with a related method that is well established and commonly used in the literature (Engbert & Kliegl, 2003).

In short, these additional analyses reinforced our central conclusions, validated our detection method, and revealed how our method was even more strict in the allocation of the critical no-microsaccade

trials – which is an important characteristic given our aims. Specifically, we have now added the following to our manuscript (note that the below is also stated in our response to the related comment 1 of reviewer 1):

Page 7 (Results):

Replication of our main findings when using another microsaccade-detection method

We replicated our main findings using an alternative, more established method for detecting microsaccades (as described in Engbert & Kliegl, 2003) that considered both horizontal and vertical displacements in gaze in both eyes. This further allowed us to validate our method and to reinforce its suitability for our aims. We report and discuss these data in **Supplementary Figure 7** and its associated supplementary discussion.

Page 24/25 (Supplementary Material):

Supplementary Figure 7. Replication of our main results using an alternative microsaccade-detection method. *a)* Time courses of gaze-shift rates (number of microsaccades per second) for shifts toward and away from the memorised location of the cued memory item. The left panel shows the results from Engbert & Kliegl’s 2003 method; the right panel shows the results from our method. *b)* The overlap of trials in each condition that sorted by Engbert & Kliegl’s method and our method. The top row represent the trials sorted by Engbert & Kliegl’s method while the bottom row represent the overlap in trial-allocation when relying on our method. *c)* In each condition, the overlays of averaged 8-12 Hz alpha lateralization from the trials that are sorted by Engbert & Kliegl’s method and by our method.

Supplementary results text associated with Supplementary Figure 7

While we titrated our method of microsaccade detection to our current aims (deliberately using a relatively low threshold for detecting, and focusing on horizontal shifts of gaze), we obtained similar results when using another, more established, method by Engbert & Kliegl (2003) instead.

First, we obtained similar temporal profiles of microsaccades and their modulation by internal selective attention (**Supplementary Fig. 7a**).

Second, we converged on strongly overlapping trial classifications when relying on our method compared to the Engbert & Kliegl method. When the Engbert & Kliegl method detected a toward microsaccade in the 200-600 ms post-cue window of interest, so did our method in the vast majority of trials (**Supplementary Fig. 7b**, left). Likewise, when the Engbert & Kliegl method detected an away microsaccade, so did our method (**Supplementary Fig. 7b**, middle). Moreover, when the Engbert & Kliegl method did not detect a microsaccade (in our window of interest), our method occasionally still detected a toward or away microsaccade (**Supplementary Fig. 7b**, right). This reveals that our method was more strict when it comes to allocating the label “no-microsaccade” to a trial (i.e. our methods seemed more liberal for allocating gaze shifts, and thereby more conservative for determining no-microsaccade trials). This is an important characteristic given our central aim of addressing whether the neural modulation is preserved in these critical no-microsaccade trials.

Finally, irrespective of these slight differences in trial-allocation between methods, alpha modulation in each of the three trial classes was highly comparable between the two microsaccade-detection methods (**Supplementary Fig. 7c**) – revealing a clear alpha modulation in the no-microsaccade trials in both cases. These data thus help validate our method, and reinforce our central conclusions by showing that our main findings generalise when using a different microsaccade-detection method.

In addition, please note how microsaccade frequency (rate in Hz) can be found in our time course plot in main Figure 1b (also depicted below for convenience). Note that the frequency of microsaccades in these plots involve microsaccade rates, separately for toward and away microsaccades. Total microsaccade frequency thus comprises the sum of the two. As can be appreciated from Figure 1b below, microsaccade rate in the neutral pre-cue period is approximately 1.4 Hz (~0.7Hz for toward plus ~0.7Hz for away). To our knowledge, this is consistent with other studies in humans.

Figure 1b.

REVIEWERS' COMMENTS

Reviewer #1 (Remarks to the Author):

The authors have addressed all my concerns. Congratulations on a very interesting and clearly written paper!

Reviewer #2 (Remarks to the Author):

The authors adequately addressed all my (minor) concerns. Moreover, seeing how they responded to the other referees' comments I would say the manuscript is now even stronger than before. Well done!

Reviewer #3 (Remarks to the Author):

Concerns were adequately addressed.